# Structural Changes and Astrocyte Response of the Lateral Geniculate Nucleus in a Ferret Model of Ocular Hypertension

**DOI:** 10.3390/ijms21041339

**Published:** 2020-02-17

**Authors:** Takashi Fujishiro, Megumi Honjo, Hiroshi Kawasaki, Ryo Asaoka, Reiko Yamagishi, Makoto Aihara

**Affiliations:** 1Department of Ophthalmology, University of Tokyo School of Medicine, Tokyo 113-8655, Japan; fujishiro-tky@umin.ac.jp (T.F.); honjomegumi@gmail.com (M.H.); ryoasa0120@googlemail.com (R.A.); kimura.reiko.0512@gmail.com (R.Y.); 2Department of Medical Neuroscience, Graduate School of Medical Sciences, Kanazawa University, Kanazawa 920-1192, Japan; hiroshi-kawasaki@umin.ac.jp

**Keywords:** LGN, ferret, glaucoma

## Abstract

We investigated structural changes and astrocyte responses of the lateral geniculate nucleus (LGN) in a ferret model of ocular hypertension (OH). In 10 ferrets, OH was induced via the injection of cultured conjunctival cells into the anterior chamber of the right eye; six normal ferrets were used as controls. Anterograde axonal tracing with cholera toxin B revealed that atrophic damage was evident in the LGN layers receiving projections from OH eyes. Immunohistochemical analysis with antibodies against NeuN, glial fibrillary acidic protein (GFAP), and Iba-1 was performed to specifically label neurons, astrocytes, and microglia in the LGN. Significantly decreased NeuN immunoreactivity and increased GFAP and Iba-1 immunoreactivities were observed in the LGN layers receiving projections from OH eyes. Interestingly, the changes in the immunoreactivities were significantly different among the LGN layers. The C layers showed more severe damage than the A and A1 layers. Secondary degenerative changes in the LGN were also observed, including neuronal damage and astrocyte reactions in each LGN layer. These results suggest that our ferret model of OH is valuable for investigating damages during the retina–brain transmission of the visual pathway in glaucoma. The vulnerability of the C layers was revealed for the first time.

## 1. Introduction

Glaucoma is a leading cause of irreversible blindness worldwide, and it is characterized by the death of retinal ganglion cells [1,2]. Additionally, glaucoma optic neuropathy (GON) leads to secondary damage to the central visual system, which processes vision via the optic nerve, the lateral geniculate nucleus (LGN), and the primary visual cortex (V1) [3,4,5,6]. In experimentally induced glaucoma in monkeys, neuronal degeneration was observed in the magnocellular, parvocellular, and koniocellular pathways in the LGN, and these changes were related to the increase in intraocular pressure (IOP) and severity of optic nerve damage [7,8]. These secondary degenerations in the central nervous system may indicate that GON is a complicated, irreversible, and progressive disease.

Anatomically, conventional rodent models, such as rats and mice, do not have the lamination of the LGN and the numbers of the LGN lamina in these models differ from those in carnivores and primates [9,10,11,12,13,14,15,16,17,18,19,20,21,22,23,24,25]. Therefore, an experimental monkey model of glaucoma has been anticipated to be an ideal model for the evaluation of central visual system changes in glaucoma. However, performing numerous experiments in monkeys is significantly challenging. Thus, we have previously reported on the development of a ferret model of ocular hypertension (OH) [26]. Ferrets possess highly developed binocular vision compared with rodents, and electrophysiological and morphological data of the ferret visual system have been accumulated [27,28,29,30,31,32,33,34,35,36,37,38,39,40].

In our ferret model of OH, damage to the optic tract projected from the OH eye was apparent and damage to the optic tract of the untreated eye was also significant [26]. Thus, secondary degeneration of the optic tract accompanied by the degeneration of the glaucomatous optic nerve may be investigated in this ferret model.

Retinal ganglion cells are functionally divided into three cell types—magno, parvo, and konio cells—in primates, including humans, and the corresponding types in ferrets are known as X, Y, and W cells [27]. Among these, konio cells have been analyzed in detail because damage to konio cells, which perform blue/yellow chromatic processing, was reported to occur earlier than that to other cell types in early stage glaucoma [4,5,41]. So far, there have been few reports regarding damage to konio cells during glaucoma pathogenesis. It seemed possible that these different susceptibilities of three types of LGN cells to trans-synaptic degeneration were caused by anatomical retinal connectivity. Another possibility is that konio cells in the LGN are more sensitive to OH than magno and parvo cells in the LGN.

The analysis of W cells in ferrets would be interesting. While konio cells project to the interlayer of the LGN in primates, the corresponding W cells project to the C layer, which clearly and independently localized in the dorsal area of the ferret LGN [4,5,30,31,33].

In this study, we aimed to analyze the changes of neurons, astrocytes, and microglia in the LGN in our ferret model of OH using immunohistochemistry.

## 2. Results

### 2.1. IOP of the OH Model

Elevation of IOP was observed in 10 ferrets from the first week after cell injection. 

In OH ferrets, at baseline, average IOP in the right and left eyes were 18.8 ± 2.0 and 17.7 ± 3.0 mmHg, respectively. At 13 weeks, Average IOP in the right (treated; G1) and left (untreated; G2) eyes was 50.0 ± 17.0 and 14.5 ± 3.9 mmHg, respectively. The IOP of treated eyes was significantly higher than the IOP of untreated eyes (*n* = 10, paired *t*-test, *p* < 0.05; Figure 1A).

In normal ferrets, IOP of untreated normal ferrets was recorded for 13 weeks. At baseline, average IOP in the right and left eyes (G3) were 15.6 ± 3.1 and 14.8 ± 3.6 mmHg, respectively in normal ferrets. At 13 weeks, average IOP in the right and left eyes were 14.9 ± 3.9 and 15.4 ± 2.4 mmHg, respectively in normal ferrets. There was no significant difference between right and left eyes in untreated normal ferrets. (*n* = 6, paired *t*-test, *p* > 0.05; Figure 1B).

### 2.2. Microscopic Analysis of the Visual Tract in the OH Model

#### 2.2.1. Histological Analysis

In normal ferrets, CTB conjugated to Alexa 555 (red label) and Alexa 488 (green label) injected into the right and left eyes, respectively, were distributed in A, A1, and C layers of both the LGNs (Figure 2A); the A layer receives contralateral afferent projections, whereas the A1 layer receives ipsilateral projections. Therefore, the A layer was stained with red CTB in the left LGN and green CTB in the right LGN, while the A1 layer was stained with green CTB in the left LGN and red CTB in the right LGN (Figure 2A). C layers contain both contralateral and ipsilateral projections (Figure 2A).

In the OH ferrets in which OH was induced only in the right eye (OH period, 13 weeks), apparently weakened CTB red projections were observed in the left A and C layers and right A1 layers of the LGN, which received afferent projections from the right OH eyes (G1). Similarly, apparently weakened CTB green projections were observed in the right A and C layers and left A1 layer of the LGN, which received afferent projections from the left untreated eyes (G2; Figure 2B).

#### 2.2.2. Quantitative Analysis of A, A1, and C Layers of the LGN in the OH Model

Quantitative analysis of the immunointensity of CTB was performed in the LGN of normal and OH ferrets (OH period, 13 weeks; normal, *n* =6; OH, *n* = 10). The intensity of CTB red in the left A, right A1, and left C layers from the right OH eyes (G1) compared with that in normal ferrets (G3) was reduced by 66.7%, 71.5%, and 80.9%, respectively (Figure 3A). The immunointensity of CTB in each LGN layers from the right OH eyes (G1) was significantly decreased compared with that in the corresponding LGN layers in the eyes of the control ferrets (G3; unpaired *t*-test, *p* < 0.01; Figure 3A).

The intensity of CTB green in the right A, left A1, and right C layers from the left untreated eyes (G2) compared with that of normal ferrets (G3) were reduced by 25.1%, 22.1%, and 26.8%, respectively (Figure 3B). The immunointensity of CTB in LGN layers from the left untreated eyes (G2) of treated ferrets was significantly decreased compared with that in the corresponding LGN layers in the eyes of control ferrets (G3; unpaired *t*-test, *p* < 0.05; Figure 3B).

### 2.3. Immunological Analysis of the Visual Tract

#### 2.3.1. Neurons

Next, we investigated the expression of NeuN in the LGN as a neuronal cell marker since it is observed in most neuronal cell types throughout the nervous system. NeuN immunoreactivity was observed in the A, A1, and C layers in both control and OH ferrets (Figure 4A).

Quantitative analysis of NeuN-immunoreactive cells in the A, A1, and C LGN layers of normal and OH ferrets (OH period, 13 weeks; normal, *n* = 6; OH, *n* = 10) was performed to elucidate the nature and extent of neuronal damage to the LGN. There was no significant difference in the numbers of NeuN-immunoreactive cells in the A, A1, and C layers of the LGN on either side in normal ferrets. In OH ferrets, significant decreases in the number of NeuN-immunoreactive cells in the A, A1, and C layers of the LGN projecting from the OH eyes (right; G1) were observed compared with those from the eyes of the control ferrets (G3; unpaired *t*-test, *p* < 0.01; Figure 4B). However, no significant decrease in the number of NeuN-immunoreactive cells in the right A, left A1, and right C layers of the LGN projecting from the untreated eyes of OH ferrets (G2), was observed compared with those from eyes of the control ferrets (G3; unpaired *t*-test, *p* > 0.05; Figure 4B).

In each LGN layer, the number of NeuN-positive cells was compared between normal and OH ferrets, and the reduction compared with normal ferrets was calculated. The number of NeuN-positive cells decreased by 36.6% in the left A, 34.6% in the right A1, and 38.4% in the left C layers of the LGN projecting from the OH eyes (G1). Decrease in the number of NeuN-positive cells was the highest in the C layer projecting from the OH eyes (G1). The number of NeuN-positive cells decreased by 6.4% in the right A, 3.5% in the left A1, and 6.6% in the right C layers of the LGN, which were projected from the contralateral untreated eyes (G2; Table 1).

#### 2.3.2. Astrocytes

GFAP-immunoreactive cells in the LGN showed small, stellate cell bodies with short, thin processes characteristic to nonreactive astrocytes. In the LGN of normal ferrets, GFAP immunoreactivity was present and evenly distributed across the three layers (Figure 5A).

There were no significant differences in GFAP immunoreactivity in the A, A1, and C layers of the LGN on either side in normal ferrets. In OH ferrets, GFAP immunoreactivity was increased and evenly distributed across all layers and reactive astrocytes were observed.

The number of GFAP-positive cells in the A, A1, and C layers of the LGN in the normal and OH ferrets were calculated to investigate glial activation in the LGN (OH period, 13 weeks; OH, *n* = 10; normal, *n* = 6). The numbers of GFAP-positive cells were increased in the A, A1, and C layers projecting from OH eyes (G1) compared with those from the untreated eyes (G2) and the normal eyes (G3). Among them, the numbers of GFAP-positive cells in the left A layer and left C layer were significantly increased (unpaired *t*-test, *p* < 0.05; Figure 5B).

Next, the number of GFAP positive cells in each LGN layer was compared between normal and OH ferrets. The number of GFAP positive cells increased by 42.3%, 6.4%, and 30.4% in the left A, right A1, and left C layers of the LGN, respectively, which were projected from the right OH eyes (G1). The number of GFAP positive cells increased by 4.9%, −3.1%, and 1.6% in the right A, left A1, and right C layers of the LGN, respectively, which were projected from the untreated eyes (G2; Table 1).

#### 2.3.3. Microglia

Iba-1 is a selective marker for microglia. In the LGN of normal ferrets, Iba-1 immunoreactivity was mainly observed in cell body and was evenly distributed across layers A, A1, and C (Figure 6A).

The number of Iba-1-positive cells in the A, A1, and C layers of the LGN in the normal and OH ferrets were calculated to investigate microglial activation in the LGN (OH period, 13 weeks; OH, *n* = 10; normal, *n* = 6). In the LGN of OH ferrets, a remarkable increase of Iba-1 immunoreactivity was observed, with strongly stained cell bodies. Iba-1-immunoreactive cells showed similar spatial and temporal distribution patterns to Iba-1 immunoreactivity in the control and glaucomatous LGN, and immunoreactivity was significantly elevated in the glaucomatous layers. Significantly increased Iba-1 immunoreactivity was observed in the A, A1, and C layers projecting from OH eyes (G1) compared with those from the untreated eyes (G2) and the normal eyes (G3; unpaired *t*-test, *p* < 0.05; Figure 6B).

The number of Iba-1-positive cells in each LGN layer was compared between normal and OH ferrets, and the rate of change compared with normal ferrets was calculated. The numbers of Iba-1-positive cells were increased by 16.2% in left A, 22.1% in right A1, and 27.5% in left C layers, which were projected from the right OH eye (G1), compared with that in the eyes of normal ferrets (G3). The numbers of Iba-1 positive cells increased by −3.2%, −2.3%, and 3.3%, respectively, in the right A, left A1, and right C layers, which were projected from the untreated eyes (G2). This increase in the number of Iba-1-positive cells was prominent in the C layer projected from the right OH eye (G1). Moreover, there was a marked decrease in the number of neurons in the C layer, suggesting that the neuronal damage and reactive glial response were the most severe in the C layer (Table 1).

## 3. Discussion

Ferrets have abundant non-cross fibers in the optic nerve, unlike mice and rats, and the C layer projected from W cells (konio cells in humans and monkeys) is easily observed as independent layers. In this study, we examined glaucomatous optic neuropathy and associated central neuropathy in the LGN using a ferret model of OH.

This is the first report of the analysis of the LGN in a ferret glaucoma model. We uncovered decreased numbers of neurons and increased activity of glial cells in the A, A1, and C layers of the LGN, projected from OH eyes. In addition, we found that the damage to the C layer was the greatest, suggesting that W cells projecting to the C layer were more sensitive and vulnerable to OH-induced neuronal damage compared with X and Y cells projecting to the A and A1 layers.

Since the LGN of mice and rats does not form a layered structure, they are not suitable animal models for analyzing glaucoma-induced CNS changes [11]. Furthermore, distinction among magno, parvo, and konio cells is unclear in mice, and therefore detailed examination cannot be conducted in rodents. In humans and monkeys, the LGN shows a six-layered structure, but the diffuse structure of konio cells across layers of the LGN makes it difficult to observe konio cells in detail [5,8]. Thus, the use of ferrets has a significant advantage in that we can observe distinct LGN layers. Similar to konio cells in monkeys and humans, W cells in the LGN project to the C layers, which exist as independent layers [30,31,33].

Our results clearly showed that damage due to OH is greater in the C layer than in the A and A1 layers, suggesting that our ferret model of OH is useful for the analysis of W cells, which correspond to konio cells in humans and monkeys.

Previous reports on central nervous system disorders in experimental animal models of OH are summarized in Table 2. In our ferret model of OH, axonal injury of the optic nerve was observed based on decreased fluorescence intensity of CTB in the LGN. Interestingly, we also found injury of axons projected from normal eyes as decreased CTB fluorescence intensity in the LGN. So far, there have been no detailed reports on bilateral LGN damage due to unilateral OH in the animal OH model. Here, we report for the first time that axonal injury from unilateral OH disturbed bilaterally in all three LGN layers. However, this model showed more acute damage than is observed in primary open angle glaucoma (POAG); therefore, it is questionable whether similar neuronal disturbances and glial activation are induced in POAG eyes [42,43,44]. A chronic OH model is required for the analysis of the secondary degeneration of CNS in POAG.

The number of NeuN-positive cells was decreased in all the layers (A, A1, and C) projected from the OH eyes (Table 1 and Table 2). Previously, decreased the number of neurons in the LGN in monkeys [7,44,45] and decreased LGN volume in humans [46] has been reported. Our findings in ferrets corroborate these results.

Astrocytes and microglia were activated in A, A1, and C layers, which were projected from the OH eyes (Figure 5B and Figure 6B). While some studies have reported that glial cells were activated during severe glaucomatous optic neuropathy [45,47,48,49,50], other did not observe this [44]. These differences may be due to differences in OH duration. In this study, because we analyzed the LGN at 13 weeks from optic nerve axonal injury due to OH, the activity of glial cells may have been retained. The relationship of IOP and the stain intensity is an interesting point, but unfortunately our study had only 13 ferrets in this experiment. The analysis of this point was difficult for this time. In order to clarify the relationship between LGN neuronal degeneration and glial activation, it would be better to observe both neuronal degeneration and glial activation as a time-course after injection of conjunctival cells. Although activated astrocytes and microglia are usually observed as cells with radial processes, such cells are not found in this experiment. In our experiments, fresh-frozen sections were used for immuno-staining because fresh-frozen sections had better reactivity to antibodies compared to sections made from perfusion-fixed tissue. In general, fine structures in fresh-frozen sections were less visible than those in perfusion-fixed sections. This seems to be the reason why it was difficult to recognize processes. In addition, we used thin 14 µm sections for immuno-staining, and it was often difficult to observe fine processes in thin sections.

There were some disadvantages of our ferret OH model. First, injection of conjunctival cells elicited intracameral inflammation, which may elicit retinal and axonal degeneration and subsequently histological changes of LGN mimicking inflammatory glaucoma unlike POAG. Second, it was hard to regulate IOP to establish a mild glaucoma model such as POAG. Thus, it may be difficult to discern if activation of astrocytes and microglia in LGN was caused by chronic IOP-dependent mimicking open angle glaucoma or subacute ischemic-damage of axons mimicking primary angle closure glaucoma. In the future, we need to overcome these disadvantages of OH ferrets models.

In conclusion, we, for the first time, examined the neuroglial responses in LGN using a ferret model of OH. Axonal injury was observed in the area of the both LGN projecting from the OH eyes. We found that W cells in the C layer, corresponding to konio cells in humans and monkeys, were greatly affected. Following a decrease in the number of neurons, glial cells were activated in LGN layers projecting from the OH eyes. Our ferret model of OH was useful for analyzing pressure-dependent CNS changes.

## 4. Materials and Methods

### 4.1. Animals

All animal experiments were performed according to the ethical guidelines for animal experimentation of the Graduate School of Medicine and Faculty of Medicine at the University of Tokyo (approval number: 1795, approval date: 10 February 2010), and all animal experiments were performed in accordance with the Guidelines for the ARVO Statement for the Use of Animals in Ophthalmic and Vision Research.

A total of 16 adult (age, 16–32 weeks) female Marshall ferrets were obtained from Marshall BioResources (New York, USA). The animals were housed at of 23 °C with a 12-h light/12-h dark cycle and were provided food and water ad libitum.

### 4.2. Preparation of Conjunctival Cells

A 1 mm × 2 mm rectangle of conjunctival tissue was excised from a ferret, minced in phosphate buffered saline with 100 µg/mL streptomycin, and cultured under standard conditions (moist atmosphere, 5 % CO_2_, 37 °C) in Dulbecco’s Minimum Essential Medium (DMEM) supplemented with 20 % fetal bovine serum (FBS), and 100 µg/mL streptomycin for 8 weeks. At this time, the cells were fully confluent and transformed into fibroblasts. Cultured fibroblasts were repeatedly cultured every 7 days after achieving full confluent.

### 4.3. Injection of Cultured Cells and Treatment of Eyes

Before the cells were fully confluent, a trypsinized cell suspension (3.3 × 10^4^ cell/mL: 50 μL) was injected into the anterior chamber of the right eye with a 32-gauge needle in 15 ferrets, and the left eye was untreated. After injection, the corneas of the injected OH right eye and non-treated left eye were treated with an ointment of 0.3 % ofloxacin.

### 4.4. Development of the Ferret Model of OH

The right eyes of 10 ferrets were established as OH eyes (G1) by injecting cultured conjunctival cells into the anterior chamber aiming the angle closure synechia [26], and the contralateral left eyes were untreated and used as a control against the right OH eye (G2). Twelve eyes of 6 ferrets with normal ocular pressure were used as controls (G3). IOP was measured every week for 13 weeks after the injection of conjunctival cells, using the TonoLab^®^ (Tiolat, Helsinki, Finland).

### 4.5. Tracer Injection

To detect the connection and degeneration of the visual system from the eye to LGN, dyes were injected as described previously [30,31]. Briefly, 5 μL Cholera toxin B (CTB) conjugated to Alexa 555 (red label) or Alexa 488 (green label) was injected into the vitreous body with a 33-gauge needle at pars plana. Red and green CTB were injected into the right and left eyes of each ferret, respectively.

### 4.6. Microscopic Analysis of the Visual Tract in the OH Model

Brains were isolated 4 days after CTB injection, embedded in optimal cutting temperature compound, and frozen. Horizontal sections including the LGN (14-μm thick) were obtained using a cryostat. Sections were fixed in 4% paraformaldehyde (PFA) in PBS for 10 min and observed using a fluorescence microscope.

To compare the average fluorescence intensities between the normal and OH ferrets, photographs of (200 × 200 μm^2^) of the A, A1, and C layers in the LGN were obtained. We obtained all of the images under the same procedure, same imaging conditions, and same contrast. In order to increase the reliability of the quantification, we used two different locations (medial and lateral locations) in layers A, A1, and C at a size of 200 × 200 μm^2^. Fluorescence intensity was measured using ImageJ^®^. The average intensity of each layer of LGN was compared between OH (*n* = 10) and normal ferrets (*n* = 6). Next, fluorescence intensities were compared between 3 LGN layers (A, A1, and C) of G1, G2, and G3, and the increase of intensities of G1 and G2 was calculated in comparison with G3.

### 4.7. Immunohistochemistry of the Visual Tract in the OH Model

Immunohistochemistry was performed as described previously with slight modifications [30,31]. In the current study, we conducted a preliminary experiment and it was confirmed that astrocytes, and microglia was stained well using the GFAP monoclonal antibody (G3893; Sigma-Aldrich, St. Louis, MO, USA) and the Iba-1 polyclonal antibody (019-19741; Wako, Osaka, Japan), respectively (Appendix A) [32].

Furthermore, these findings were observed in G3. Immunostaining was performed under an identical same procedure, same imaging condition, and same contrast condition, and the procedures were processed with the same protocol by the same person, we used the same antibody lot and the same immunohistochemistry was done in the same day. Horizontal sections (14-μm thick) were made using a cryostat, fixed in 4% PFA for 10 min, permeabilized with 0.1% Triton X-100/PBS, and washed with 0.01 M PBS (pH 7.4). They were then pre-incubated with 10% normal goat serum in 0.01 M PBS for 30 min and incubated for 1 day at 4 °C with primary antibodies, which included mouse anti-NeuN monoclonal antibody (1:1000 dilution; MAB377; Chemicon, USA), mouse anti-glial fibrillary acidic protein (GFAP) monoclonal antibody (1:600 dilution) (G3893; Sigma, USA), and rabbit anti-ionized calcium-binding adaptor molecule 1 (Iba-1) polyclonal antibody (1: 600 dilution; 019-19741; Wako, Osaka, Japan) in a solution of 10% normal goat serum in 0.01 M PBS containing 0.3% Triton X-100. Horizontal sections were then washed with PBS and incubated with biotinylated anti-mouse or anti-rabbit IgG before being incubated with avidin–biotin–peroxidase complex for 30 min at room temperature. Finally, diaminobenzidine was used as a peroxidase substrate for visualization.

### 4.8. Quantification

To quantify NeuN, GFAP, and Iba-1-positive cells, a photograph of (200 × 200 μm^2^) of the A, A1, and C layers of the LGN was obtained. In order to increase the reliability of the quantification, we use two different locations (medial and lateral locations) in layers A, A1, and C at a size of 200 × 200 μm^2^. The numbers of NeuN, GFAP, and Iba-1-positive cells were counted using ImageJ^®^.

### 4.9. Statistics

All data are shown as mean and standard deviation. Differences in IOP between injected and control eyes were statistically evaluated using a paired *t*-test with Bonferroni correction. Difference in the numbers of the NeuN- and Iba-1-positive cells in the A, A1, and C layers of the LGN between the injected and control eyes were evaluated using a paired *t*-test. Difference in the intensity of GFAP staining in the A, A1, and C layers of the LGN between the injected and control eyes were evaluated using a paired *t*-test.

## Figures and Tables

**Figure 1 ijms-21-01339-f001:**
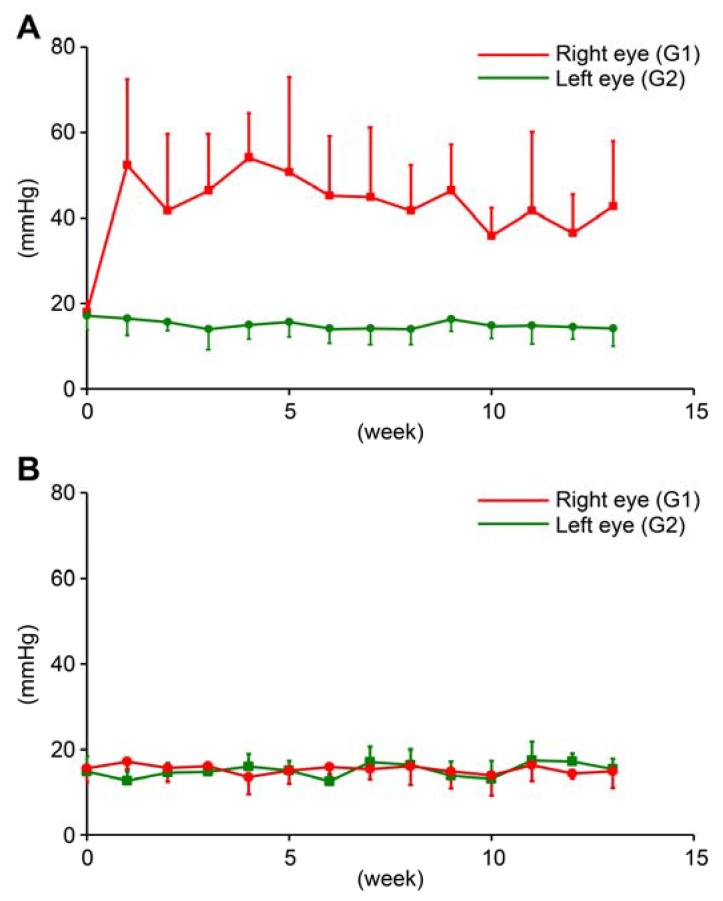
Intraocular pressure (IOP) of ocular hypertension (OH) eyes (G1), contralateral untreated eyes of OH ferret (G2), and normal eyes of control ferret (G3). (**A**): Red line indicates IOP of right OH eyes, and green line indicates IOP of contralateral left untreated eyes in OH ferret (mean ± S.D.). (**B**): Red and green line indicate IOP of right and left eyes in control ferret (mean ± S.D.).

**Figure 2 ijms-21-01339-f002:**
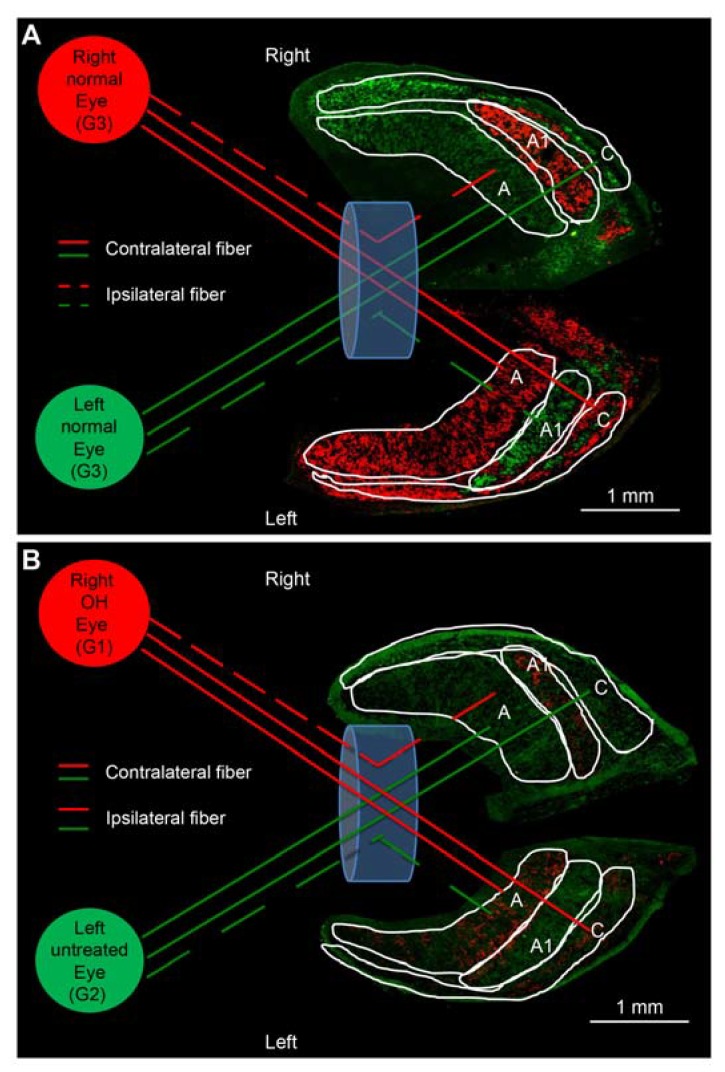
Axonal projection from the eye to lateral geniculate nucleus (LGN) layers in the normal and OH ferret. (**A**): A and C layers project to the contralateral side, whereas A1 layer projects to the ipsilateral side. Red and green CTB injected into the right and left eyes, respectively, are projected on both sides of LGN in control ferrets and looks similar staining intensity (white bar = 1 mm). (**B**): In OH ferrets, left A and C layers and right A1 layers of LGN projecting from the right OH eye (G1) showed weaker CTB staining intensity compared to those from the left untreated contralateral eyes (G2; white bar = 1 mm).

**Figure 3 ijms-21-01339-f003:**
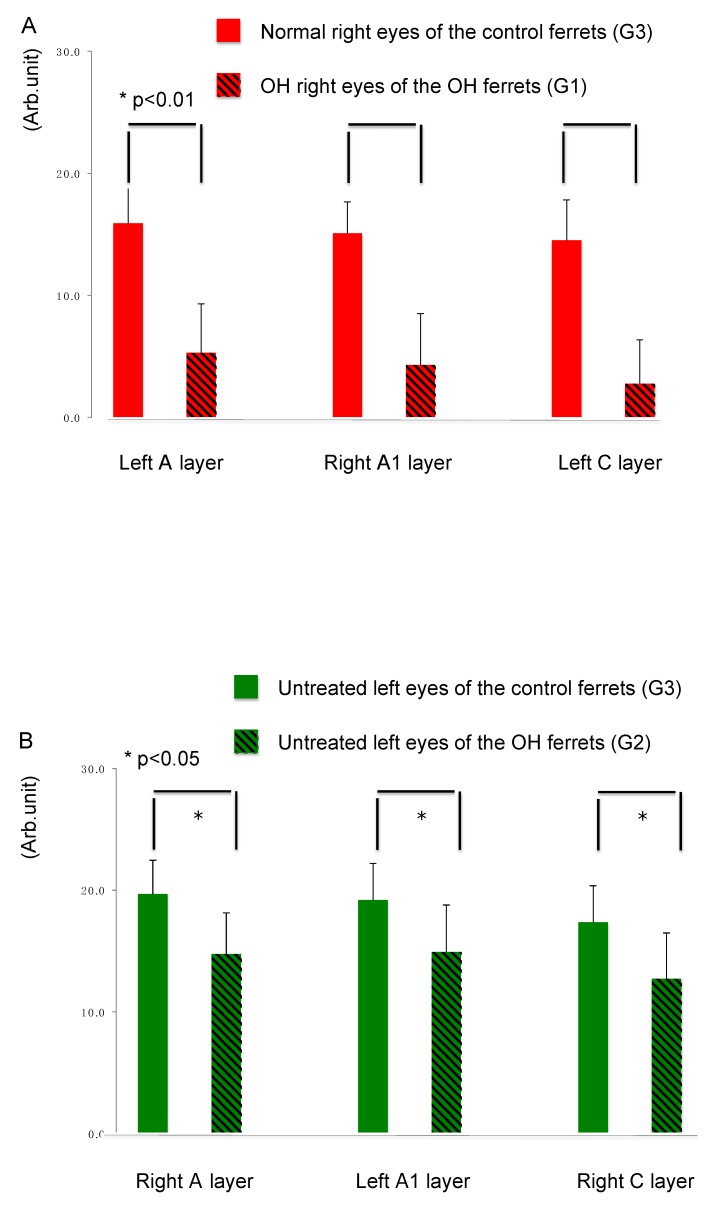
Comparison of axon damage in LGN layers of the OH ferret and the control ferret. (**A**): The immunointensity of CTB in each LGN layer projected from the OH right eyes (G1) of OH ferrets was significantly decreased than in corresponding LGN layers in the untreated right eyes of control ferrets (G3; *n* = 10, unpaired *t*-test, *p* < 0.01). (**B**): The immunointensity of CTB in each LGN layer projected from the contralateral untreated left eyes (G2) of OH ferrets was significantly decreased than in corresponding LGN layers in the untreated left eyes of control ferrets (G3; *n* = 10, unpaired *t*-test, *p* < 0.05).

**Figure 4 ijms-21-01339-f004:**
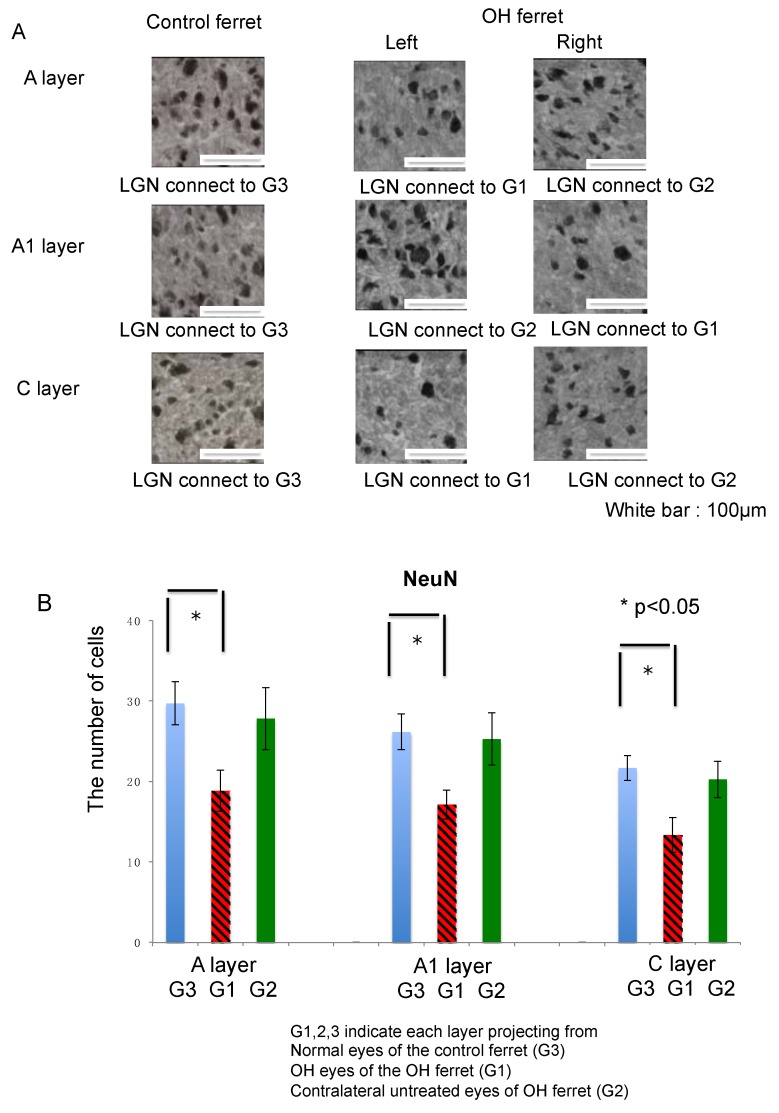
Immunohistological images of neurons in LGN of the OH (G1) and the untreated contralateral eyes (G2) of the OH ferret, and the normal eyes (G3) of the control ferret. (**A**): NeuN staining indicating neuronal cells. (200 μm × 200 μm square; white bar = 100 μm). (**B**) Comparison of neurons in LGN layers projecting from the OH (G1) and the untreated contralateral eyes (G2) of the OH ferret, and the normal eyes (G3) of the control ferret. B: The number of NeuN immunoreactive neuronal cells. Significant decrease of the number of NeuN-immunoreactive cells was observed in A, A1, and C layers of G1, compared with those layers of G3 (*n* = 10, paired *t*-test, *p* < 0.01).

**Figure 5 ijms-21-01339-f005:**
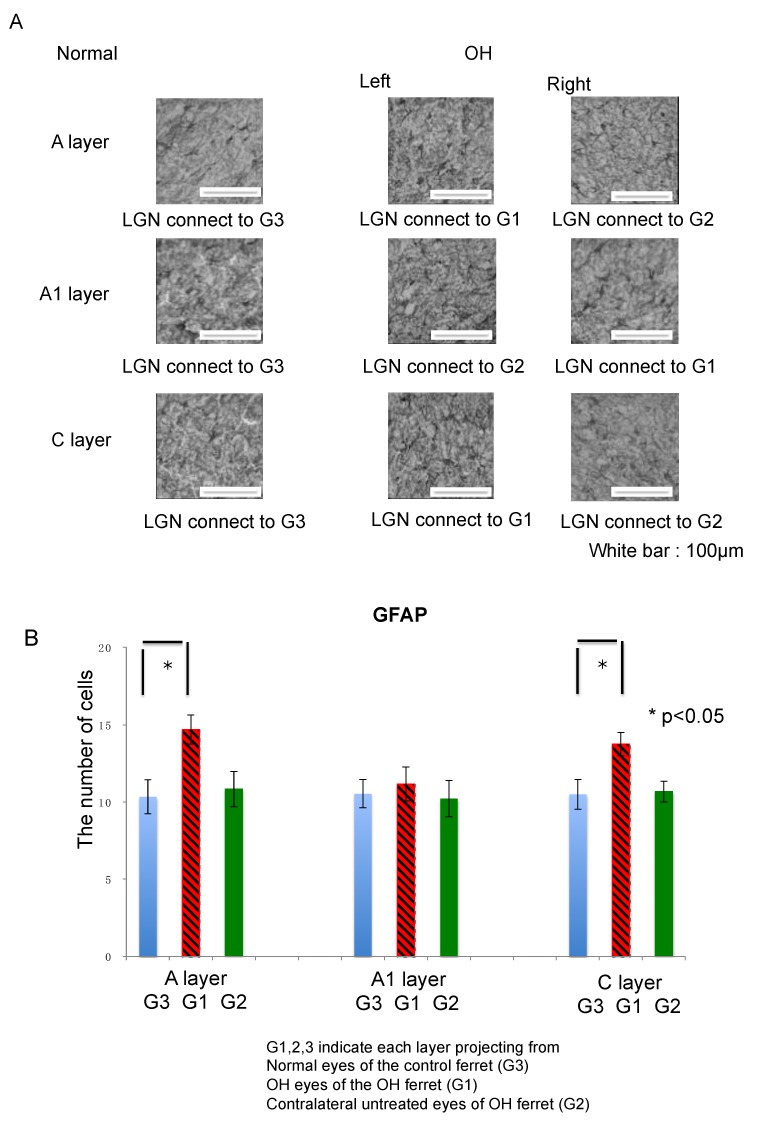
Immunohistological images of neuroglial cells in LGN of the OH (G1) and the untreated contralateral eyes (G2) of the OH ferret, and the normal eyes (G3) of the control ferret. (**A**): GFAP staining indicating microglia. (200 μm × 200 μm square; white bar = 100 μm). Comparison of neuroglial cells in LGN layers projecting from the OH (G1) and the untreated contralateral eyes (G2) of the OH ferret, and the normal eyes (G3) of the control ferret. (**B**): The number of GFAP immunoreactive glial cells. Significant increase of the number of GFAP-1-immunoreactive cells were observed in the A and C layers of G1, compared with those of G3 (*n* = 10, paired *t*-test, *p* < 0.05).

**Figure 6 ijms-21-01339-f006:**
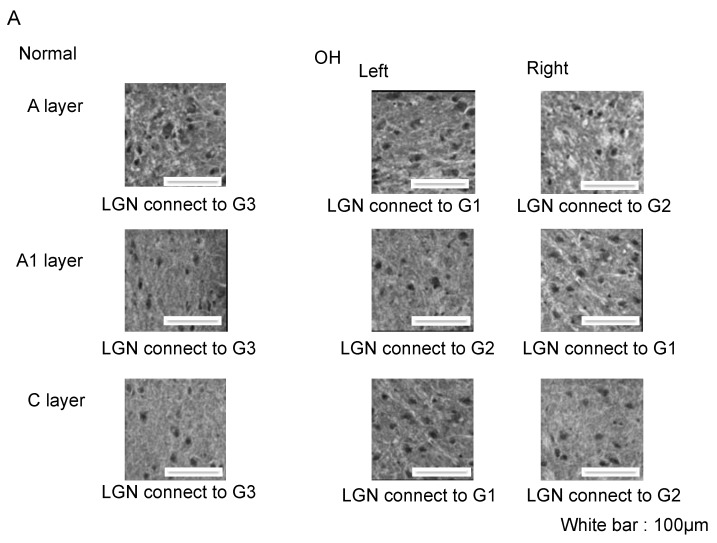
Immunohistological images of neuroglial cells in LGN of the OH (G1) and the untreated contralateral eyes (G2) of the OH ferret, and the normal eyes (G3) of the control ferret. (**A**): Iba-1 staining indicating microglia. (200 μm × 200 μm square; white bar = 100 μm). Comparison of neuroglial cells in LGN layers projecting from the OH (G1) and the untreated contralateral eyes (G2) of the OH ferret, and the normal eyes (G3) of the control ferret. (**B**): The number of Iba-1 immunoreactive microglial cells. A significant increase of the number of Iba-1-immunoreactive cells was observed in A, A1, and C layers of G1, compared with those of G3 (*n* = 10, paired *t*-test, *p* < 0.05).

**Table 1 ijms-21-01339-t001:** Quantitative analysis of projecting axons and neuroglial responses in A, A1, and C layers of OH ferret LGN.

Target	Labeling Method	Red (Elevated IOP)			Green (Normal IOP)		
		A	A1	C	A	A1	C
Axon	CTB	−66.79%	−71.5%	−80.9%	−25.1%	−22.1%	−26.8%
Neuron	NeuN	−36.6%	−34.6%	−38.4%	−6.4%	−3.5%	−6.6%
Astrocyte	GFAP	42.3%	6.4%	30.4%	4.9%	−3.1%	1.6%
Microglia	Iba-1	16.2%	22.1%	27.5%	−3.2%	−2.3%	3.3%

The intensity of CTB red and green in each LGN layers indicate projecting axons from OH eyes (G1) and contralateral untreated eyes (G2) of OH ferrets compared with those from control eyes (G3). The number of NeuN-positive cells, GFAP immuno-intensity, the number of Iba-1-positive cells represent neurons, astroglia, and microglia in each LGN layers. The numbers indicate the increased rate of immuno-intensity of layers projecting from OH eyes (G1) and contralateral untreated eyes (G2) of OH ferrets compared to the normal eyes of the control ferrets (G3). CTB: Cholera toxin B; NeuN: Neuronal nuclei; GFAP: Glial fibrillary acidic protein; Iba-1: Ionized calcium binding adapter molecule 1; A: A layer of LGN; A1: A1 layer of LGN; C: C layer of LGN.

**Table 2 ijms-21-01339-t002:** Comparison of LGN damage induced by OH in ferrets, monkeys, and rats.

Target	Labeling Method	Ferret	Monkey27, 28	Rat31, 33–36
IOP (Elevated IOP duration)		13 w	6 m to 1 y	14 d to 8 m
Axon activity	CTB	Bilateral damage	−	−
Neuron	NeuN	A, A1, and C layers damage	damage	damage
Astrocyte	GFAP	Only A layer damage	activation	activation
Microglia	Iba-1	A, A1, and C layers activation	activation	activation

LGN: lateral geniculate nucleus; OH: ocular hypertension; CTB: Cholera toxin B; NeuN: Neuronal nuclei; GFAP: Glial fibrillary acidic protein; Iba-1: Ionized calcium binding adapter molecule 1; A: A layer of LGN; A1: A1 layer of LGN; C: C layer of LGN.

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
