# Peer review of "Structural Changes and Astrocyte Response of the Lateral Geniculate Nucleus in a Ferret Model of Ocular Hypertension"

_ijms, 2020, doi:10.3390/ijms21041339_

Round 1
Reviewer 1 Report
In this study, the authors observed alterations in neurons and glial cells in the LGN in a glaucoma model using ferrets whose optic nervous system is close to that of human. It was found that astrocytes and microglia were activated at the LGN regions where the damaged optic nerves project. Activation of glial cells is known to be involved in many neurodegenerative diseases, and it seems to be meaningful to examine the relationship with degeneration of LGN neurons in glaucoma. However, the histochemical images of LGN, which are the main data of this study, are unclear and not convincing. Also, the methods of quantifying histochemical images are not appropriate. Therefore, it is difficult to evaluate the significance from these results.
The histochemical images of DAB staining in Fig. 4, which are major findings in this study, are unclear and does not seem to show the changes described by the authors. In particular, the stainings of GFAP-positive astrocytes and Iba1-positive microglia in Fig. 4B and C do not show the typical morphology. Although activated astrocytes and microglia are usually observed as cells with radial processes, such cells are not found in Fig. 4B and C. These histochemical observations need to be reexamined. As for antibodies used in Fig. 4, the providers of the GFAP monoclonal antibody (G3893; Sigma) and the Iba-1 polyclonal antibody (019-19741; Wako) do not validate their immunoreactivities to ferret. Therefore, the authors need to show that these antibodies have immunoreactivities against ferret GFAP and Iba-1. The authors should describe the methods to quantify the immunobiological image in detail. In the Materials and Methods, it is described that cells in one image of 200 x 200 microm square were counted. However, the area seems to be small compared to the size of LGN regions shown in Figure 2. In order to increase the reliability of the quantification, a larger area needs to be observed, such as counting multiple photographs obtained from the same ferret. Also, the numbers of NeuN positive cells and the Iba-1 positive cells shown in Fig. 4A do not agree with the cell numbers shown in Fig. 5A and C. I cannot find 20-40 positive cells in Fig. 4A and C. In addition, as for quantification of GFAP-positive cells, staining intensity, but not counting cell numbers, was used (Fig. 5B). The authors should label GFAP-positive cells more clearly, and count the cell numbers as was done in NeuN and Iba-1-positive cells. As shown in Table 2, it has been reported that neuronal degeneration and glial activation in the LGN occur in glaucoma models. However, the causal relationship between neuronal degeneration and glial activation is not clear. In this study, these changes were observed only at 13 weeks after injection of cultured conjunctival cells. In order to clarify the relationship between LGN neuronal degeneration and glial activation, it would be better to observe both neuronal degeneration and glial activation as a time-course after injection of conjunctival cells.
Author Response
We would like to thank the editor and the reviewers for their constructive and insightful comments. As suggested, we modified and improved our manuscript intensively. Our responses to the comments were written below.
Reviewer 1
Comments and Suggestions for Authors
In this study, the authors observed alterations in neurons and glial cells in the LGN in a glaucoma model using ferrets whose optic nervous system is close to that of human. It was found that astrocytes and microglia were activated at the LGN regions where the damaged optic nerves project. Activation of glial cells is known to be involved in many neurodegenerative diseases, and it seems to be meaningful to examine the relationship with degeneration of LGN neurons in glaucoma. However, the histochemical images of LGN, which are the main data of this study, are unclear and not convincing. Also, the methods of quantifying histochemical images are not appropriate. Therefore, it is difficult to evaluate the significance from these results.
Thank you very much for reviewing our manuscript very carefully. Please see our comments below.
The histochemical images of DAB staining in Fig. 4, which are major findings in this study, are unclear and does not seem to show the changes described by the authors.
Thank you very much for this insightful comment. We agree with the reviewer that the qualities of images were not good enough. We therefore replaced the photographs (Fig. 5a). We agree with the reviewer that it is difficult to recognize the difference of the staining from the images at a glance. Therefore, we conducted quantitative investigations as shown in Figure 4b, 5b and 6b. As a result, significant decrease of NeuN, increase of GFAP and increase of Iba-1 were observed in OH-treated eyes compared to control and untreated eyes.
In particular, the stainings of GFAP-positive astrocytes and Iba1-positive microglia in Fig. 4B and C do not show the typical morphology. Although activated astrocytes and microglia are usually observed as cells with radial processes, such cells are not found in Fig. 4B and C. These histochemical observations need to be reexamined.
We totally agree with the reviewer on this point. In our experiments, fresh-frozen sections were used for immune-staining because fresh-frozen sections had better reactivity to antibodies compared to sections made from perfusion-fixed tissue. In general, fine structures in fresh-frozen sections are less visible than those in perfusion-fixed sections. This seems to be the reason why it was difficult to recognize processes. In addition, we used thin 14 µm sections for immuno-staining, and it is often difficult to observe fine processes in thin sections. We considered this is an important point of this experiment and therefore added a comment to the Discussion section (Page 12, Lines 1-8).
(Page 12, Lines 1-8)
Although activated astrocytes and microglia are usually observed as cells with radial processes, such cells are not found in this experiment. In our experiments, fresh-frozen sections were used for immuno-staining because fresh-frozen sections had better reactivity to antibodies compared to sections made from perfusion-fixed tissue. In general, fine structures in fresh-frozen sections are less visible than those in perfusion-fixed sections. This seems to be the reason why it was difficult to recognize processes. In addition, we used thin 14 µm sections for immuno-staining, and it is often difficult to observe fine processes in thin sections.
As for antibodies used in Fig. 4, the providers of the GFAP monoclonal antibody (G3893; Sigma) and the Iba-1 polyclonal antibody (019-19741; Wako) do not validate their immunoreactivities to ferret. Therefore, the authors need to show that these antibodies have immunoreactivities against ferret GFAP and Iba-1. The authors should describe the methods to quantify the immunobiological image in detail.
We thank the reviewer to raise this point. As suggested, there have been no GFAP monoclonal and Iba-1 polyclonal antibodies specific to ferret. We have conducted a preliminary confirmation experiment and confirmed that astrocytes, and microglia was stained well using the GFAP monoclonal antibody (G3893; Sigma) and the Iba-1 polyclonal antibody (019-19741; Wako), respectively. We added the reference (ref 32) that confirms their immunoreactivities. Furthermore, these findings were observed in normal (control) LGN. Thus, we are certain that these antibodies have immune-reactivities against ferret GFAP and Iba-1 in ferret. We have revised the manuscript in the Method section to explain this issue (Page 15 Line 8-12). We also added photographs of negative control in supplemental data.
Ref 32
Mizuguchi K, Horiike T, Matsumoto N, Ichikawa Y, Shinmyo Y & Kawasaki H. Distribution and Morphological Features of Microglia in the Developing Cerebral Cortex of Gyrencephalic Mammals. Neurochem Res. 43, 1075-1085 (2018).
(Page 15 Line 8-12)
In the current study, we conducted a preliminary experiment and it was confirmed that astrocytes, and microglia was stained well using the GFAP monoclonal antibody (G3893; Sigma, USA) and the Iba-1 polyclonal antibody (019-19741; Wako, Osaka, Japan), respectively (supplemental data)32.
In the Materials and Methods, it is described that cells in one image of 200 x 200 microm square were counted. However, the area seems to be small compared to the size of LGN regions shown in Figure 2. In order to increase the reliability of the quantification, a larger area needs to be observed, such as counting multiple photographs obtained from the same ferret.
Thank you for the reviewer's comment. It is possible to consider getting images of larger areas in A layer. However, the thicknesses of Layer A1 and C are only 200 μm at most, so that we obtained the images of limited area of 200 x 200 microm square. As suggested by the reviewer, we used two different locations (medial and lateral locations) in layers A, A1, and C at a size of 200 x 200 μm2 each in order to increase reliability of the quantification in the revised manuscript. We described this point in the Method section (Page 14, Lines 23-24 to Page 15, Line 1-2 ; Page 16, Lines 7-9). We replaced all graphs and tables with those using new calculation results.
(Page 14, Lines 23-24 to Page 15, Lines 1-2)
We obtained all of images under a same procedure, same imaging conditions and same contrast. In order to increase the reliability of the quantification, we use two different locations (medial and lateral locations) in layers A, A1, and C at a size of 200 x 200 μm2.
(Page 16, Lines 7-9)
In order to increase the reliability of the quantification, we use two different locations (medial and lateral locations) in layers A, A1, and C at a size of 200 x 200 μm2.
Also, the numbers of NeuN positive cells and the Iba-1 positive cells shown in Fig. 4A do not agree with the cell numbers shown in Fig. 5A and C. I cannot find 20-40 positive cells in Fig. 4A and C.
Thank you very much. As described in our response above, we replaced these images to avoid confusion. The results in the current study are based on the outcome of the quantitative investigation (Figure 4a, 5a, 6a).
In addition, as for quantification of GFAP-positive cells, staining intensity, but not counting cell numbers, was used (Fig. 5B). The authors should label GFAP-positive cells more clearly, and count the cell numbers as was done in NeuN and Iba-1-positive cells.
As suggested by the reviewer, we counted the numbers of GFAP-positive cells in the new photographs as was done for NeuN- and Iba-1-positive cells. These results were shown as new Figure 5a and 5b and were written in the text (Page 16, Lines 9-10).
(Page 16, Lines 9-10)
The numbers of NeuN, GFAP and Iba-1-positive cells were counted using ImageJ®.
As shown in Table 2, it has been reported that neuronal degeneration and glial activation in the LGN occur in glaucoma models. However, the causal relationship between neuronal degeneration and glial activation is not clear. In this study, these changes were observed only at 13 weeks after injection of cultured conjunctival cells. In order to clarify the relationship between LGN neuronal degeneration and glial activation, it would be better to observe both neuronal degeneration and glial activation as a time-course after injection of conjunctival cells.
We are grateful for this insightful comment. We agree with the reviewer, and this point was added in the text (Page 11, Lines 21-24 to Page 12, Line 1). Not a few previous studies have reported that neuronal degeneration and glial activation occur in the LGN in glaucoma models (ref 5). Consistent with this, we observed similar changes in the LGN 13 weeks after the injection of cultured conjunctival cells. In order to further clarify the causal relationship between neuronal degeneration and glial activation in the LGN, it would be needed to investigate a time-course of neuronal degeneration and glial activation after the injection of conjunctival cells. This is beyond the scope of the current study, and we are planning to perform such investigation in our future study.
(Page 11, Lines 21-24 to Page 12, Line 1)
The relationship of IOP and the stain intensity is interesting point, but un-fortunately our study has only 13 ferrets in this experiment. The analysis of this point is difficult for this time. In order to clarify the relationship between LGN neuronal degeneration and glial activation, it would be better to observe both neuronal degeneration and glial activation as a time-course after injection of conjunctival cells.
Reviewer 2 Report
This study was designed with the aim to study changes in neurons, astrocytes, and microglia of the lateral geniculate nucleus in a ferret model of ocular hypertension.
This is an immunohistochemistry study which provide a valuable methods usefull to investigate pressure-dependent CNS changes during glaucoma.
Experimental design is well done and the results are clearly presented.
Author Response
We would like to thank the editor and the reviewers for their constructive and insightful comments. As suggested, we modified and improved our manuscript intensively. Our responses to the comments were written below.
Reviewer2
Comments and Suggestions for Authors
This study was designed with the aim to study changes in neurons, astrocytes, and microglia of the lateral geniculate nucleus in a ferret model of ocular hypertension.
This is an immunohistochemistry study which provide a valuable methods usefull to investigate pressure-dependent CNS changes during glaucoma.
Experimental design is well done and the results are clearly presented.
Thank you very much for your insightful review and positive comments.
Reviewer 3 Report
This is an interesting manuscript describing morphological changes in the lateral geniculate nucleus after induction of oculasr hypertension. The article can be improved by
Methods: describe the measurements of average fluorescence intensities. Have you normalized relative fluorescence intensities in photographs or used real readings? How did you quantify the intensity of GFAP stainings between different measurements/photographs? Besides the effects of rising the intraocular opressure, can affect absolute intensities ion a photograph. How did you deal with this issue? Results: Presentation is not optimal. As an example - under the subtitle "Neurons" there are figures representing the effect of increased ocular pressure on neurons, astrocytes and micorglia. There are no figures under subtitles "Astrocytes" and "Microglia". It might be better to reduce the number of subtitles and put figures on more appropriate places in the manuscript.Author Response
Comments and Suggestions for Authors
This is an interesting manuscript describing morphological changes in the lateral geniculate nucleus after induction of oculasr hypertension. The article can be improved by
Methods: describe the measurements of average fluorescence intensities. Have you normalized relative fluorescence intensities in photographs or used real readings?
Thank you very much for this comment. We obtained all of images under the same procedure, same imaging conditions and same contrast. We added this information in the text (Page 14, Lines 23-24).
(Page 14, Lines 23-24)
We obtained all of images under a same procedure, same imaging conditions and same contrast.
How did you quantify the intensity of GFAP stainings between different measurements/photographs? Besides the effects of rising the intraocular opressure, can affect absolute intensities ion a photograph. How did you deal with this issue?
Thank you very much for your comments. To minimize the difference between distinct GFAP-related measurements/photographs, GFAP staining was performed under the same procedure, same imaging condition and same contrast condition. Hence the influence of this issue would be negligible. We added this information in the Method section (Page 15, Lines 12-13).
We agree with the reviewer. The relationship of IOP and the stain intensity is an interesting point. Unfortunately, however, our study used only 13 ferrets in this experiment, and the analysis of this point was difficult for this time. We would investigate this point in our future experiments. Thank you for your valuable opinion. Because we think this is an important point, we wrote this point in the Discussion section (Page 11, Lines 21-24 to Page 12, Line 1).
(Page 15, Lines 12-13)
Furthermore, these findings were observed in G3. Immuno-staining was performed under an identical same procedure, same imaging condition and same contrast condition,
(Page 11, Lines 21-24 to Page 12, Line 1)
The relationship of IOP and the stain intensity is interesting point, but un-fortunately our study has only 13 ferrets in this experiment. The analysis of this point is difficult for this time. In order to clarify the relationship between LGN neuronal degeneration and glial activation, it would be better to observe both neuronal degeneration and glial activation as a time-course after injection of conjunctival cells.
Results: Presentation is not optimal. As an example - under the subtitle "Neurons" there are figures representing the effect of increased ocular pressure on neurons, astrocytes and micorglia. There are no figures under subtitles "Astrocytes" and "Microglia". It might be better to reduce the number of subtitles and put figures on more appropriate places in the manuscript.
Thank you for your advice and we apologize for our mistakes and confusion. We modified "Astrocytes" and "Microglia" sections so that "Astrocytes" section and "Microglia" section have own figures and graphs, and the order of figures was changed accordingly.
Round 2
Reviewer 1 Report
None.